# Computer Vision-Based Measurement Techniques for Livestock Body Dimension and Weight: A Review

Weihong Ma [1,†], Xiangyu Qi [1,2,†], Yi Sun [1], Ronghua Gao [1], Luyu Ding [1], Rong Wang [1], Cheng Peng [1], Jun Zhang [1], Jianwei Wu [1], Zhankang Xu [1], Mingyu Li [1], Hongyan Zhao [3], Shudong Huang [1,4,*] and Qifeng Li [1,*]

1. Information Technology Research Center, Beijing Academy of Agriculture and Forestry Sciences, Beijing 100097, China; mawh@nercita.org.cn (W.M.); qxy@cau.edu.cn (X.Q.); sunyi@nwafu.edu.cn (Y.S.); gaorh@nercita.org.cn (R.G.); dingly@nercita.org.cn (L.D.); rongw@nwafu.edu.cn (R.W.); pengc@nercita.org.cn (C.P.); zhangj@nercita.org.cn (J.Z.); wujw@nercita.org.cn (J.W.); 1043283468@nwafu.edu.cn (Z.X.); limy@nercita.org.cn (M.L.)
2. College of Information and Electrical Engineering, China Agricultural University, Beijing 100083, China
3. Otoke Banner Agricultural and Animal Husbandry Technology Extension Center, Ordos 016199, China; etkqnmjstgzx@163.com
4. College of Computer Science, Sichuan University, Chengdu 610065, China
* Correspondence: huangsd@scu.edu.cn (S.H.); liqf@nercita.org.cn (Q.L.)
† These authors contributed equally to this work.

**Abstract:** Acquiring phenotypic data from livestock constitutes a crucial yet cumbersome phase in the breeding process. Traditionally, obtaining livestock phenotypic data primarily involves manual, on-body measurement methods. This approach not only requires extensive labor but also induces stress on animals, which leads to potential economic losses. Presently, the integration of next-generation Artificial Intelligence (AI), visual processing, intelligent sensing, multimodal fusion processing, and robotic technology is increasingly prevalent in livestock farming. The advantages of these technologies lie in their rapidity and efficiency, coupled with their capability to acquire livestock data in a non-contact manner. Based on this, we provide a comprehensive summary and analysis of the primary advanced technologies employed in the non-contact acquisition of livestock phenotypic data. This review focuses on visual and AI-related techniques, including 3D reconstruction technology, body dimension acquisition techniques, and live animal weight estimation. We introduce the development of livestock 3D reconstruction technology and compare the methods of obtaining 3D point cloud data of livestock through RGB cameras, laser scanning, and 3D cameras. Subsequently, we explore body size calculation methods and compare the advantages and disadvantages of RGB image calculation methods and 3D point cloud body size calculation methods. Furthermore, we also compare and analyze weight estimation methods of linear regression and neural networks. Finally, we discuss the challenges and future trends of non-contact livestock phenotypic data acquisition. Through emerging technologies like next-generation AI and computer vision, the acquisition, analysis, and management of livestock phenotypic data are poised for rapid advancement.

**Keywords:** 3D reconstruction; stressless body dimension measurement; visual weight estimation; precision livestock farming

## 1. Introduction

Livestock husbandry has long been an integral component of the agricultural sector, significantly impacting food supply, rural economies, and environmental sustainability. However, with the ongoing global population increase, there is a noticeable surge in demand for high-quality animal protein, a trend that cannot be disregarded [1]. In 2003, research related to Precision Livestock Farming (PLF) was first compiled during the European Conference on Precision Livestock Farming. The conference primarily focused on animal physiological identification [2–4] and monitoring [5,6]. The aim was to optimize

individual animal contributions, achieving efficiency in livestock farming at low costs and environmental footprints, while ensuring the quality and safety of livestock products [7]. Consequently, PLF, amid environmental changes [8], resource scarcity [9], and inadequate agricultural labor [10], is poised to emerge as a significant trend and developmental direction in the field of livestock farming [11].

Phenotypic data of animals form a fundamental basis in PLF for various aspects such as breeding management [12], genetic selection [13], scientific research [14], and more. Only by accumulating extensive data throughout the livestock growth process and establishing comprehensive databases covering the entire production process can scientific research methods be employed effectively for optimizing and enhancing breeding production decisions. This ensures product quality, protects genetic breeding stock, and provides a robust foundation for sustainable development and innovation within the livestock industry [15].

Typically, phenotypic data on growth include livestock body height, length, and weight, among other measurements. Presently, the predominant method for collecting livestock phenotypic data mainly involves manual acquisition. This method requires individuals to use devices such as rulers and weighing scales for data collection. However, livestock control is challenging and often necessitates the forceful restriction of their movements [16]. This approach demands substantial manual labor, consumes time and labor resources, and can induce stress responses in livestock. For instance, Zulkifli mentioned that human contact measurement lowers the productivity of farm animals, leading to reduced fertility, milk production, and growth rates [17]. In the process of manually observing and measuring key areas in livestock, different observers might have their own subjective views, which can lead to inconsistent decisions in how they gather data. This inconsistency might result in errors within the collected data, ultimately reducing its reliability.

In recent years, the use of intelligent sensing devices for data collection has emerged as a promising solution to mitigate the problems associated with manual data collection. For instance, deploying devices like infrared sensors [18–20], 3D point cloud cameras [21,22], and RGB cameras [23,24] in farming or animal research settings facilitates the capture of livestock images and 3D coordinate points on their surfaces. Subsequently, employing 3D reconstruction techniques generates complete 3D models of livestock. Following this, computer vision methods automatically or semi-automatically identify key points of the animals, ultimately computing phenotypic data such as body measurements and weight. The workflow of non-contact 3D reconstruction and data estimation for animals is illustrated in Figure 1.

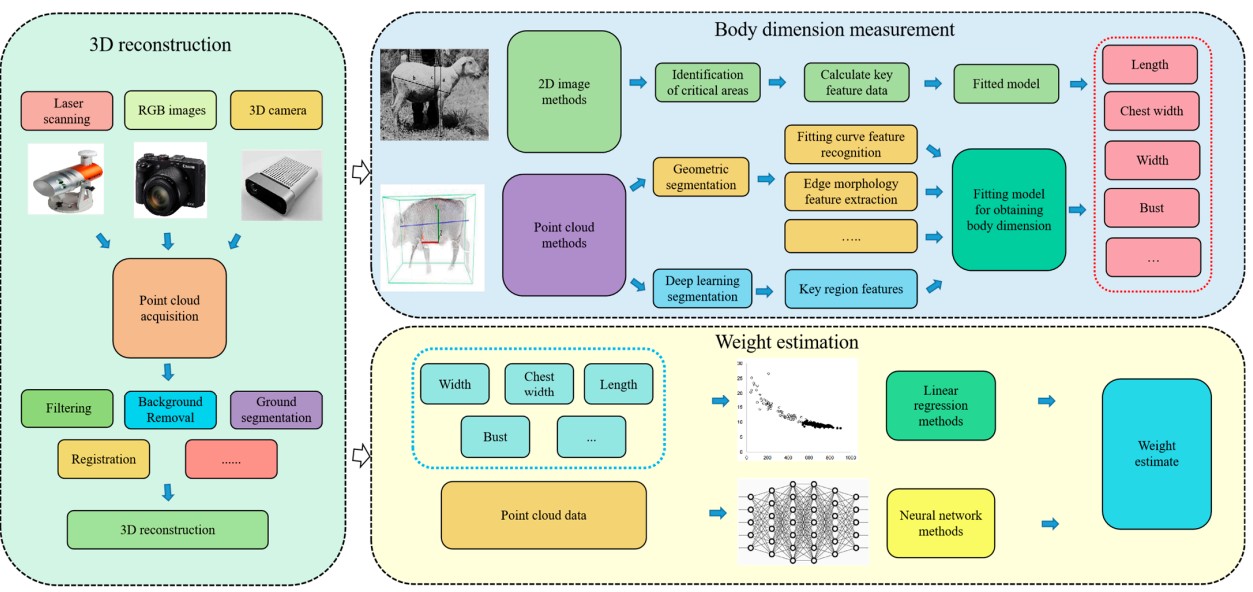

**Figure 1.** Computer vision-based phenotypic data acquisition technical framework.

Using intelligent sensing devices to acquire animal phenotypic data eliminates the need for direct contact with animals, reducing potential stress induced by manual collection methods. Simultaneously, this approach allows for the collection of vast amounts of data in a short period, proving more efficient than manual methods [25]. This is particularly advantageous for large-scale farming operations and research projects [26]. Undoubtedly, this method represents a potential solution capable of enhancing data collection efficiency, accuracy, and animal welfare.

In this paper, our focus lies on non-contact intelligent sensing technology for livestock. Specifically, we emphasize intelligent perception and analysis techniques related to three primary tasks: (1) computer vision-based 3D reconstruction of livestock; (2) computer vision-based livestock body dimension acquisition technology; and (3) computer vision-based livestock weight estimation technology. Within this work, we summarize and analyze the latest advancements in these fields and discuss future research opportunities, as well as the associated challenges.

## 2. Computer Vision-Based Livestock 3D Reconstruction Technology

The significance of 3D reconstruction technology within PLF cannot be overstated. This method facilitates the provision of comprehensive and accurate digital models of animals, allowing for the acquisition of extensive livestock phenotypic data, thereby establishing a crucial foundation for informed decision-making. For instance, by monitoring and analyzing animal phenotypic data, breeders can proactively devise feeding schedules [27] and detect potential issues at an earlier stage [28], ultimately enhancing production quality and minimizing resource wastage. The integration of 3D reconstruction technology serves as a vital tool for intelligent management in animal husbandry, propelling the livestock industry toward elevated levels of advancement. Thus, 3D reconstruction technology is indispensable in PLF. Presently, three primary categories of this technology are known: reconstruction based on laser scanning, reconstruction based on RGB images, and reconstruction based on 3D cameras, as depicted in Figure 2, with associated research findings outlined in Table 1.

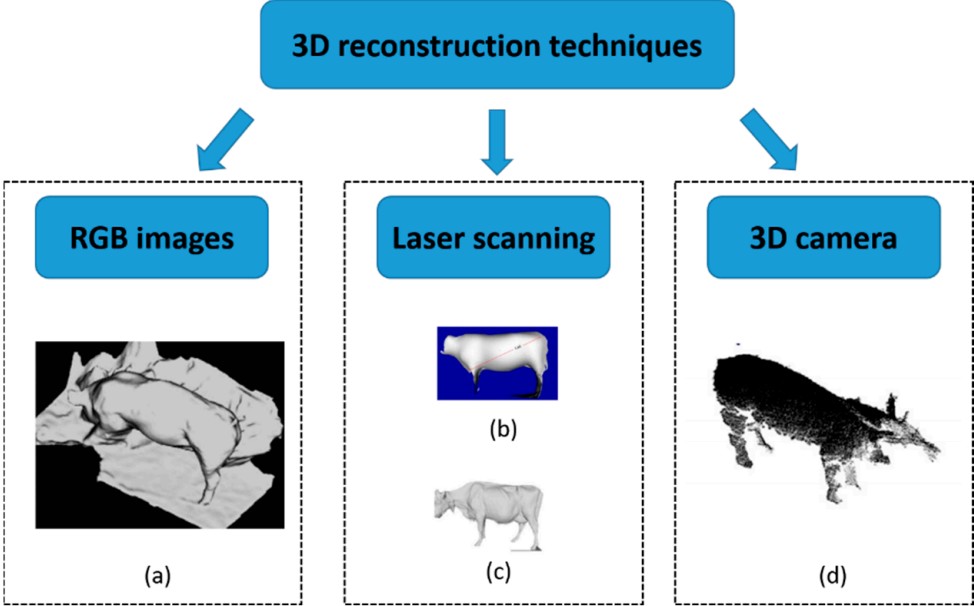

**Figure 2.** Three different 3D reconstruction methods based on RGB images, laser scanning, and 3D cameras. (**a**) displays a 3D reconstruction technique utilizing RGB images [29]; (**b**,**c**) shows two different 3D reconstruction methods based on laser scanning [30,31]; (**d**) is reconstructed using a 3D camera [32]. These techniques provide innovative computer vision methods for precise, non-contact measurement of livestock body dimensions and weight.

**Table 1.** Overview of main 3D reconstruction research in livestock.

| Work | Breed | Device | Method | Animal Numbers | Year |
|------|-------|--------|--------|----------------|------|
| [29] | Live pigs | RGB camera | Binocular stereo vision technology | 32 | 2004 |
| [30] | Live cows | LiDAR | Statistical outliers and voxel grid filtering methods | 3 | 2018 |
| [31] | Live cows | LiDAR | Fusion | 30 | 2019 |
| [32] | Live pigs | 3D camera | Point cloud registration | 20 | 2018 |
| [33] | Newborn lambs | RGB camera | Digital image processing | 158 | 2015 |
| [34] | Live sheep | RGB camera | Binocular stereo vision technology | 27 | 2014 |
| [35] | Live cows | LiDAR | Image fusion | 25 | 2023 |
| [36] | Live pigs | 3D camera | Point cloud registration | 78 | 2018 |
| [37] | Live cows | 3D camera | Point cloud registration | 101 | 2016 |
| [38] | Live pigs | Visible image and infrared image sensor | Multi-source image fusion | N/A | 2020 |

The "N/A" in [38] we don't find the number of pigs.

### 2.1. 3D Reconstruction of Livestock Based on RGB Images

The reconstruction technique based on RGB images generally involves employing a multi-source image fusion approach to acquire the target's geometric shape and texture information [39]. Its procedure typically entails the following steps: (1) capture a set of two-dimensional images containing the target object from various angles; (2) identify and extract features such as key points, corners, edges, etc. from these images, matching the features extracted from different images to determine their corresponding relationships in 3D space; and (3) leverage known camera parameters and feature matching results to achieve 3D reconstruction of the target.

For instance, Thapar et al. [40] captured images of live pigs using a smartphone from both top and side angles to gather phenotypic information. Khojastehkey [33], on the other hand, utilized a digital camera to capture side-view images of newborn lambs, followed by applying digital image processing and measurement techniques to assess the lambs' body size. Menesatti et al. [34] captured images of sheep using two network cameras positioned at different angles and later employed a binocular stereovision system to gather information about the sheep. Wu et al. [29] developed a 3D reconstruction system comprising six cameras, employing stereovision techniques to achieve 3D reconstruction of pigs. Pezzuolo [41] proposed a Structure from Motion (SfM) photogrammetric method, enabling the 3D reconstruction of pigs. Their results indicated that after capturing and utilizing 50 photographs, the reconstructed area reached 60%.

However, these RGB-based reconstruction methods face significant limitations due to the absence of a third dimension, potential distortions, the necessity for calibration procedures, and the requirement for multiple cameras. As presented in [41], the reconstructed area only reached 60%, which obviously does not meet the requirements for subsequent body size and weight calculations. Consequently, their effectiveness has been greatly constrained.

### 2.2. Livestock 3D Reconstruction Based on Laser Scanning

Laser scanning-based 3D reconstruction technology is an advanced method extensively applied in various fields such as geographic information systems, robot navigation, autonomous driving, and industrial manufacturing. This technique operates on the principle of utilizing laser radar equipment to emit laser beams and measure their return time to acquire 3D spatial information about the target object. Some scholars have already begun exploring the application of this technology in the measurement of livestock bodies.

For instance, Huang et al. [30] utilized a LiDAR (Light Detection and Ranging) sensor to collect original point cloud data of live cattle. They subsequently applied fusion conditions, statistical outlier removal, and voxel grid filtering methods to eliminate background noise and outliers. Then, employing bidirectional random K-D tree accelerated

iterative closest point (ICP) matching and greedy projection triangulation (GPT) reconstruction methods, they reconstructed the surface of cattle, ultimately obtaining the 3D model. Cozler et al. [31], on the other hand, employed a gantry structure equipped with five cameras and five laser projectors and termed their approach Morpho3D. During the data collection process, the gantry structure passed over the cow at a speed of 0.5 m/s, as depicted in Figure 3a. Subsequently, the corresponding cameras captured laser stripe images projected onto the cow's body, transmitting these data to a computer for reconstructing the cow's 3D information. Morpho3D successfully achieved high-precision 3D reconstruction of cows, demonstrating experimental results with an average distance error of less than 1 cm. Additionally, Los et al. [42] utilized a UAV (Unmanned Aerial Vehicle) equipped with a laser scanner to collect 3D point cloud data of a herd of cattle and extract individual beef cattle, achieving 3D reconstruction of the cattle as depicted in Figure 3b.

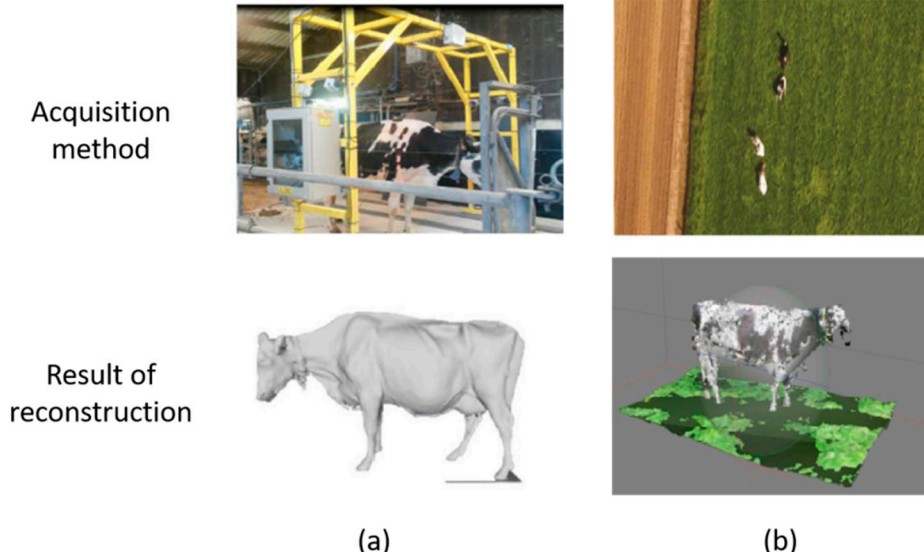

**Figure 3.** Acquisition method and the result of 3D reconstruction in [31,42], (**a**) is the result of 3D reconstruction in [31], (**b**) is the result of 3D reconstruction in [42].

However, these methods have their drawbacks. Huang et al.'s [30] approach encounters difficulties in obtaining the complete contour of the target due to the target's mobility, leading to erroneous registration and reconstruction. The method of Los [42] involves expensive equipment costs, and the UAV flight process faces several uncertainties such as the movement of cattle, adverse weather conditions like rainfall, and high wind speeds, hindering the implementation's certainty. Additionally, Morpho3D requires approximately 6 s for a single cattle capture and a reconstruction duration of 15 min, necessitating the cow to remain still during data collection, thereby limiting Morpho3D's practicality.

In summary, the methods utilizing laser scanning for 3D reconstruction face limitations stemming from target mobility and prohibitively expensive equipment, contributing to their diminished reliability in practical applications.

### 2.3. Depth-Camera-Based 3D Reconstruction

In recent years, with technological advancements, 3D cameras have garnered attention among researchers in the field of 3D reconstruction due to their superior depth information acquisition, high real-time capabilities, and comparatively lower cost compared with laser scanning [35,43–45]. These cameras capture both depth and color images of a scene simultaneously, subsequently transforming the depth images into point clouds to accomplish 3D reconstruction.

For instance, Pezzuolo et al. [46] placed two Kinect v1 cameras in a feeding area. They scanned pigs 5–10 times during feeding, selecting the best scan based on minimum noise principles. This allowed for the collection of 3D information from the side and top views of

the pigs. Subsequently, through calibration and filtering, they achieved a 3D reconstruction of the pigs, as illustrated in Figure 4b. In another study, Pezzuolo et al. [47] utilized four different scanning positions to reconstruct cattle in three dimensions. Spoliansky et al. [48] employed Kinect v1 to gather depth information from cows, estimating the body condition score based on denoised depth images. Wang et al. [49] designed a point cloud acquisition system based on the Xtion camera, capturing point clouds of pig scenarios from two different perspectives. They utilized Random Sample Consensus (RANSAC) to eliminate background point clouds and employed Euclidean clustering to extract foreground point clouds, ultimately achieving a 3D reconstruction of pigs, as shown in Figure 4a.

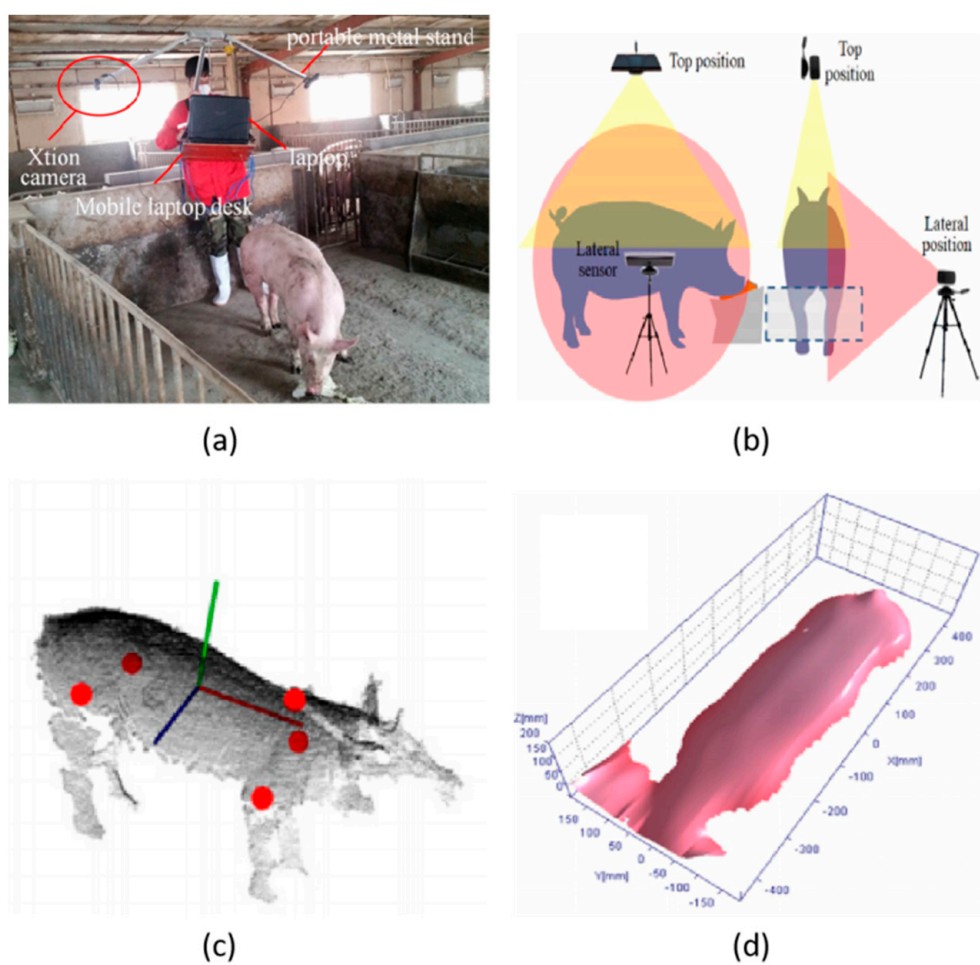

**Figure 4.** Depth camera-based 3D reconstruction methods. (**a**,**b**) The scenes where the point clouds are acquired in [46,49], respectively. (**c**,**d**) The 3D reconstruction methods in [46,49], respectively.

### 2.4. Summary of 3D Reconstruction Technology

Livestock 3D reconstruction is a prerequisite for obtaining phenotypic data such as body dimensions and weight estimation, and it plays a crucial role in aiding livestock managers in formulating scientifically informed decisions. Traditional RGB image fusion methods not only exhibit slower detection speeds but also demand stringent reconstruction environments, making them less practical for actual livestock production settings. Laser scanning reconstruction methods are better suited for extracting specific contours/parts, necessitating precise alignment between animals and sensor sources, and are generally accompanied by higher costs for laser scanning equipment [36].

Compared with traditional approaches, 3D camera-based reconstruction methods have emerged as a promising research direction. This approach directly acquires depth information, reducing feature matching complexities and making data acquisition less intricate. Additionally, 3D cameras enable real-time information capture, making them more

suitable for practical production applications, thereby mitigating the impact of livestock movement. Pezzuolo et al. [50] discussed the impact of Structure from Motion (SfM) photogrammetry methods, low-cost laser scanning, and the depth camera on 3D reconstruction. Through comparative experiments, they concluded that utilizing the Microsoft Kinect v1 3D camera for reconstruction was the most cost-effective technology.

However, despite the evident advantages of 3D cameras in certain applications, they also come with limitations and constraints that need consideration when selecting and employing this technology. Notably, 3D cameras are typically more expensive than traditional RGB cameras, including the initial cost of purchasing the 3D camera itself and potential additional equipment and software, rendering them less economically viable in certain applications. Some 3D cameras are sensitive to lighting conditions; intense light, reflections, or shadows may result in depth discontinuities and grainy noise, affecting the reconstruction of 3D models [50]. The reconstruction of data collected using 3D cameras might be more intricate than RGB images due to the complexity of the livestock environment, resulting in numerous data interference points and thereby limiting the effectiveness of existing point cloud registration methods. A comparative analysis of these three methods is presented in Table 2.

**Table 2.** Comparison of 3D reconstruction techniques with different devices.

| Device | Measurement Range | Measure Targets | Advantage | Disadvantage |
|---|---|---|---|---|
| 3D camera | From several centimeters to several meters | Acquiring spatial coordinate information of an object | Provide depth information and high-resolution depth information | Limited measurement range, susceptible to lighting and environmental conditions, higher costs, and rather intricate 3D reconstruction |
| Laser scanning | Ranges from several meters to several kilometers | Generate high-precision geometric surface point clouds of objects | Obtain highly accurate depth information of the measured object, with relatively minor susceptibility to lighting effects | Lack of color information, lower resolution, and relatively higher cost |
| RGB camera | Between tens of centimeters to several meters | Using electronic sensors to convert optical images into electronic data | Lower cost, well-developed application scenarios | Lack of depth information and susceptible to lighting and environmental conditions |

Future research endeavors could explore the impacts of different lighting conditions and potential remedies, such as developing advanced algorithms to identify and rectify issues like light, reflections, and shadows. Moreover, researchers may delve into novel point cloud registration methods coupled with neural network techniques, representing a potential trend in the point cloud registration domain. Finally, as existing studies face limitations due to animal movement, the future direction of 3D reconstruction should aim to reduce scanning time and determine optimal data acquisition positions.

## 3. Computer Vision-Based Livestock Body Dimension Acquisition Technology

The traditional method of collecting body size parameters in livestock involves farm personnel using tape measures, enclosures, and other tools to measure the pigs in the pigsty. With the introduction of PLF and the rapid advancement in computer vision technology capable of processing two-dimensional or 3D image data, researchers have progressively embarked on exploring non-contact body measurement techniques. Compared with conventional livestock surface parameter acquisition methods, non-contact body measurement technology eliminates the influence of human factors, delivering measurement results in a short duration, and significantly enhancing measurement efficiency and accuracy [37]. Presently, non-contact body measurement technology [51] can be categorized into two types: image-based methods and 3D point cloud methods, as shown in Figure 5 and Table 3.

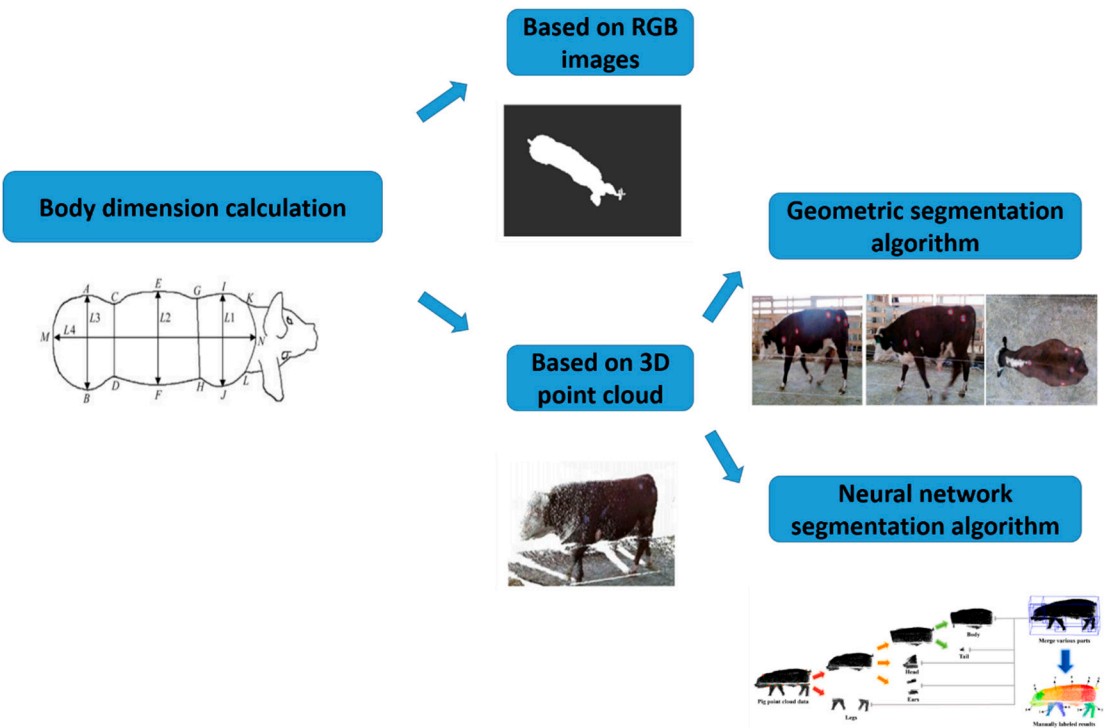

**Figure 5.** Livestock body dimension acquisition technology.

**Table 3.** Main research work of body dimension acquisition.

| Device Type | Breed | Data Type | Method | Error | Reference | Year |
|---|---|---|---|---|---|---|
| RGB camera | Live pigs | RGB images | Area method | Less than 2.94% | [32] | 2018 |
| | Live pigs | RGB images | Image processing | Less than 10.2% | [37] | 2014 |
| 3D camera | Live cows | Point cloud | Geometric segmentation | Less than 3% | [52] | 2020 |
| | Live pigs | Video | Geometric segmentation | Less than 4.5% | [53] | 2019 |
| | Live pigs | Point cloud | Geometric segmentation | Less than 7.87% | [54] | 2019 |
| | Live pigs | Point cloud | Geometric segmentation | Less than 4.67% | [55] | 2020 |
| | Live pigs | Point cloud | Neural network segmentation | Less than 5.26% | [56] | 2023 |
| | Live cows | Point cloud | Geometric segmentation | Less than 3.47% | [57] | 2023 |

*3.1. RGB Image-Based Body Dimension Measurement Method*

In recent decades, 2D cameras have been widely adopted in the field of computer vision due to their cost-effectiveness and efficiency. Numerous researchers have proposed various algorithms to extract livestock body dimensions from 2D images. Lu et al. [32] presented an algorithm to automatically extract porcine body parameters from overhead view images of pigs. The process involves the following steps: firstly, the parameters of the pig's spinal bone are extracted; secondly, the length of line segments perpendicular to the pig's skeletal line are calculated, and then feature points along the pig's contour are extracted based on the variation in these perpendicular line segment lengths; thirdly, the pig's head and neck are eliminated from its contour using an ellipse; and finally, four length parameters and one area parameter are computed. The accuracy of body length reached 95.12% (SE = 2.66%). Weber et al. [52] developed software for calculating beef cattle rump width, body length, and shoulder width based on manually labeled images, measuring the width of the beef cattle's rump and back area via image analysis. Shi et al. [53] developed a mobile measurement system for estimating pig body composition, capturing back images of growing pigs using a binocular camera, and processing the images to estimate pig body dimensions. Chen et al. [58] designed a mobile measurement system based on the platform, comprising control, calibration, and image processing sections. The calibration part is

used to identify distortion coefficients and compute camera parameters. After calibration, a binocular camera captures images of pigs during growth, and subsequent processing estimates the pig's body parameters. The system verified body length and height with $R^2$ values between 0.91 and 0.98. Sakir et al. [59] employed digital images to determine key body measurements like height, hip height, and length of Holstein cattle, achieving an accuracy of around 98%.

However, methods relying on 2D images face challenges including susceptibility to variations in illumination, difficulties in background separation, and high demands on image capture environments [60]. As shown in [61], the error reached 10.2%. Two-dimensional image-based information cannot measure three-dimensional body parameters such as chest circumference, abdominal circumference, and hip circumference, limiting the precision of livestock body measurements.

### 3.2. 3D Point Cloud Body Dimension Measurement Method

With the emergence of 3D cameras due to their exceptional spatial information acquisition capabilities and higher real-time performance, scientists have increasingly applied them to capture and analyze the 3D point cloud structure of livestock. Using a 3D camera to acquire a large amount of discrete point coordinate data reflects the spatial information of the object's surface, providing more information compared with two-dimensional images. Following the 3D reconstruction, segmentation of the reconstructed livestock point cloud is fundamental for obtaining body size phenotypic data. Currently, there are two segmentation methods: one relies on geometric segmentation algorithms, while the other utilizes segmentation models based on neural network networks [62,63]. It is worth noting that regardless of the method employed, the objective remains consistent—to segment points, lines, and surfaces from the complete point cloud of livestock and localize key regions. After segmenting points, lines, and surfaces from the livestock point cloud, relevant knowledge in animal science can be used to fit or train models to obtain the body dimension phenotypic data of the livestock.

### 3.2.1. Geometry-Based 3D Point Cloud Segmentation

Currently, the main method for segmentation involves cutting the complete point cloud based on the morphological features of livestock. These key points and regions are usually determined by geometric segmentation methods such as identifying fitted curve features and extracting edge morphology features. This category of methods has been studied for many years, and the technology has become relatively mature, as shown in Figure 6.

For instance, Yongshen et al. [54] used the minimum bounding rectangle to adjust the posture of pigs and employed projection and background difference methods to detect targets. They combined skeletonization algorithms with Hough transform algorithms to determine the tilt degree, achieving ideal posture detection and body measurements of pigs. Experimental results showed average accuracies of 95.5%, 96.3%, and 97.3% for body width, height, and length, respectively. Ruchay et al. [64] utilized the ICP algorithm to perform non-rigid 3D reconstruction of point cloud data captured by three Kinect V2 cameras on live cattle. This led to the calculation of nine body measurement parameters, including shoulder height, hip height, and oblique length, with measurement errors below 3%.

Wang et al. [65] used cross-sectional features of point clouds to detect the measurement position of the pig's heart circumference. Slicing the point cloud at the measurement position resulted in a heart circumference point cloud. Finally, the perimeter of the fitted heart circumference point cloud curve provided the heart circumference length, demonstrating an average relative error of 7.87% for the pig's heart circumference measurement position. Shuai et al. [55] obtained point clouds of freely walking pigs using a Kinect depth camera from three different perspectives (top view, left view, and right view). Through the curve of the difference in point cloud point distribution, they identified key positions of the pig's body point cloud. Additionally, they improved the accuracy of abdominal circumference

measurements using polar coordinate transformation methods. Guo et al. [66] identified key points for pig body measurements using features such as the position of pig hooves and arc length curvature, employing methods like peak detection and pole value positioning.

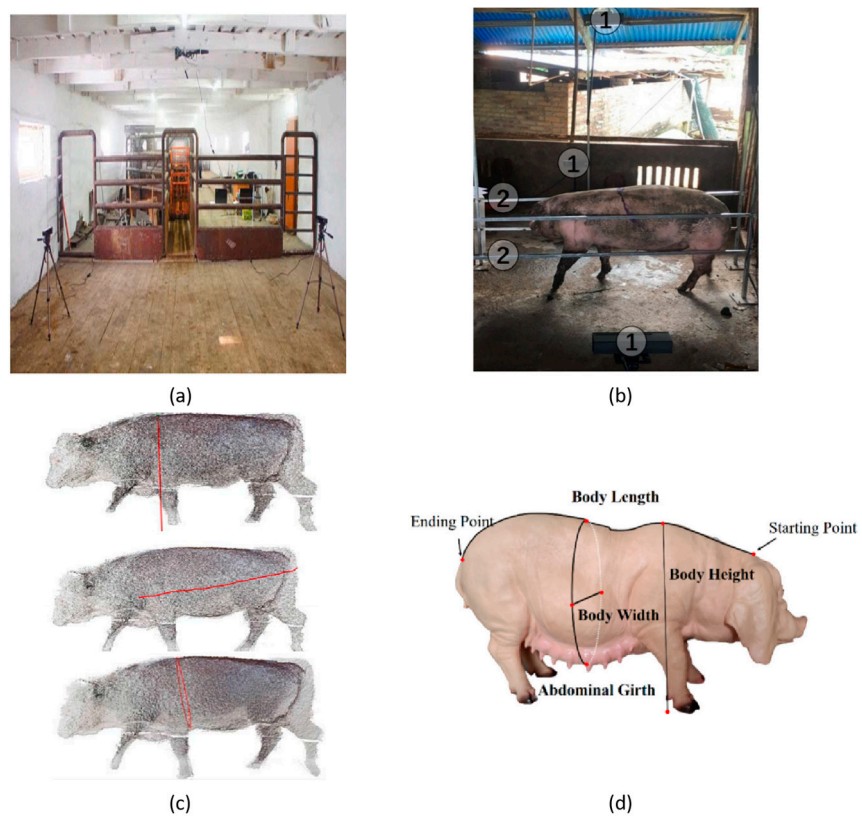

**Figure 6.** The geometric segmentation methods based on RGB images. (**a**,**b**) The scenarios of acquiring point cloud data in [55,65], respectively. (**c**,**d**) The key point detection in [55,65], respectively.

Although algorithms based on geometric segmentation can precisely segment the key areas and points in livestock point clouds, most experiments are conducted under strict fixed conditions and environments. In different production environments, variations in the posture of livestock targeted by 3D cameras may result in incorrect localization of livestock, leading to inaccurate body size measurements. Additionally, individual differences among captured livestock can cause different key points to be located differently. Employing a uniform segmentation algorithm for diverse individuals will reduce the accuracy of the algorithm and result in significant errors.

### 3.2.2. Neural Network-Based 3D Point Cloud Segmentation

To achieve precise segmentation of livestock point clouds and identify accurate key areas, researchers have explored the application of neural network methods in livestock point cloud segmentation. Specifically, researchers manually label various key areas of the livestock to form a dataset. This dataset is then used to train a 3D convolutional neural network to learn features of different key areas in the livestock. With a sufficient amount of data, the trained neural network model can achieve accurate region segmentation for different individuals.

Hao et al. [56] proposed an improved point cloud segmentation model that subdivides the overall pig point cloud into various parts, such as the pig's head, ears, torso, limbs, and tail, and localizes body measurement key points in segmented areas, as shown in Figure 7. Subsequently, Hao et al. combined algorithms like least squares, point cloud slicing, edge extraction, and polynomial fitting to achieve pig body dimension measurement. The relative error in experimental results ranged between 2.18% to 5.26%. Du et al. [67]

introduced a deep learning network-based plant segmentation network (PST) that achieved semantic segmentation of high-resolution rapeseed plant point clouds. This model comprises a Dynamic Voxel Feature Encoder (DVFE), dual attention blocks, and a dense feature propagation module, enhancing automatic segmentation capabilities for rapeseed plant point clouds. Experimental results indicated an overall semantic segmentation accuracy of 97.07%, demonstrating the significant potential of deep learning-based point cloud segmentation methods in handling dense plant point clouds with complex morphological features. Guo et al. [68] proposed a segmentation method for cabbage point cloud data by combining deep learning and clustering algorithms. This approach optimized the workflow of the DBSCAN algorithm and exhibited strong performance in organ-level plant point cloud segmentation experiments, achieving an accuracy of 95% and an IoU of 0.86. Li et al. [69] developed a deep learning-based plant point cloud segmentation technique using PointNet for stem and leaf instance segmentation, extracting six phenotypes. The results showed an average precision of 0.91 for stem–leaf segmentation, offering a systematic reference for automated analysis of 3D phenotypic features at the individual plant level.

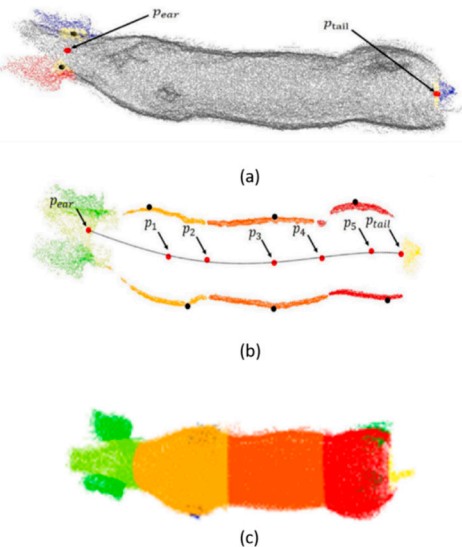

**Figure 7.** Neural network-based key area segmentation [56]. Subfigure (**a**) shows the determined ear–root point pairs and tail–root point pairs. Subfigure (**b**) depicts a schematic of the backline in a planar projection. Subfigure (**c**) presents the results after segmenting the pig's body.

However, using neural network networks for point cloud segmentation currently has limitations. This is because deep convolutional networks require a large volume of point cloud data to learn features of key areas in the point cloud. However, there is currently a scarcity of publicly available livestock point cloud datasets, making it challenging to establish datasets with sufficient data volume. Consequently, neural networks are prone to overfitting during the training process, resulting in limited practicality of the models.

### 3.3. Summary of Body Dimension Acquisition Technology

The introduction of PLF has accelerated the development of non-contact body measurement technologies. Over the past decade, researchers have conducted extensive studies in this field. While methods based on two-dimensional images have enabled the acquisition of livestock body measurements, they lack essential depth information, resulting in insufficient accuracy. Consequently, researchers have begun exploring methods based on 3D point clouds to obtain body measurement information. This is because point cloud data contain abundant spatial information, compensating for the shortcomings of two-dimensional image information.

Segmenting livestock point clouds to identify crucial areas for body measurement involves two approaches: geometric segmentation and neural network methods. Geometric

segmentation accurately segments the key areas of individual livestock, but its robustness is affected by differences between animal individuals, making it less practical for real-world applications. Although neural network methods reduce the impact of individual differences, their development might be constrained by insufficient data currently available. A comparative analysis of methods is shown in Table 4.

**Table 4.** Comparison of body dimensions calculated with different methods.

| Methods | Advantage | Disadvantage |
|---|---|---|
| RGB Image | Lower cost, more mature image processing techniques, and simpler algorithms | Lack of depth information, restricted by viewing angle |
| Geometric segmentation methods in 3D imaging | Requires minimal annotated data, high computational efficiency, and superior segmentation precision | Highly dependent on geometric accuracy, susceptible to individual animal variations |
| Neural network methods in 3D imaging | Adapts to intricate information and structures, learns crucial insights | Large data requirements, limited public datasets, and relatively high computing resource demands exist |

In the future, researchers could explore capturing and sharing more livestock point cloud data. Additionally, improving existing neural network structures to better suit the segmentation of key areas in livestock point clouds is a direction for future development.

## 4. Computer Vision-Based Livestock Weight Estimation Technology

Currently, the majority of farms and research institutions acquire livestock weight information primarily using electronic devices such as weigh crates and floor scales. While this method provides the most accurate weight measurements, it is time-consuming and may cause stress or harm to the livestock. In recent years, several scholars have conducted numerous studies on non-contact weight estimation, as shown in Table 5. These studies mainly fall into two categories: weight estimation based on mapping the relationship between 3D reconstructions and body weight, and weight estimation using deep convolutional neural networks.

**Table 5.** Main research work on livestock weight estimation.

| Breed | Device | Data Type | Method | Error | Reference | Time |
|---|---|---|---|---|---|---|
| Live pigs | 3D camera | Point cloud | Linear regression | 4.87% | [21] | 2021 |
| Live pigs | 3D camera | Point cloud | Linear regression | 0.48 kg | [41] | 2018 |
| Live pigs | 3D camera | Point cloud | Linear regression | 2.961 kg | [51] | 2022 |
| Live pigs | 3D camera | 3D images and 2D images | Neural network | MAE = 6.366 | [61] | 2021 |
| Live cows | 3D scanning device | 3D images | Linear regression | 20–30 kg | [70] | 2019 |
| Live cows | 3D scanning device | 3D images | Linear regression | 9.7% | [71] | 2022 |
| Live cows | RGB camera, infrared rangefinder | RGB images | Linear regression | LBW = 86.3–97.2% | [72] | 2019 |
| Live pigs | 3D camera | 3D images and 2D images | Neural network | 1.16 kg | [73] | 2021 |
| Live pigs | 3D camera | Point cloud | Neural network | MAE = 9.25, RMSE = 12.3 kg | [74] | 2023 |

### 4.1. Linear Regression Weight Estimation-Based Body Weight Measurement

Linear regression-based weight estimation from body measurements involves obtaining target body dimensions through 3D reconstruction of point cloud data and establishing a linear regression model based on the relationship between these dimensions and the weight of the animal [75].

Cozler et al. [70] developed a regression model utilizing parameters like volume and area obtained through 3D reconstruction of point clouds to estimate cow weights. Experimental results indicated prediction errors of 0.3% (20 kg). In a subsequent study by Cozler et al. [71], they adopted a different approach by estimating weight based on volume. Initially, they used Morpho3D with five cameras to generate 3D reconstructions of cattle. Then, employing the Poisson surface reconstruction method from the point cloud, they constructed a triangular mesh to determine volume and subsequently computed heart girth (HG), chest depth (CD), hip width (HW), and hip width (KW) using the software. Finally, they estimated weight using regression models, confirming a strong correlation between volume and weight. Yan et al. [72] established a multiple regression equation for estimating yak weight based on wither height (WH), body diagonal length (BDL), and body side area (BSA). Okayama et al. [21] estimated pig weight using a simple regression model based on "adjusted volume", achieving MAPE and RMSPE of 4.87% and 6.13%, respectively. Li et al. [76] used three regression analysis models—stepwise regression, ridge regression, and partial least squares regression—to estimate pig body measurements. They established a stepwise regression model with body length, body height, and shoulder width as independent variables and weight measured on a scale as the dependent variable, achieving an MAE of 2.961 kg.

The aforementioned studies have achieved the estimation of livestock weight. However, their accuracy is relatively low. For instance, the study referenced as [71] indicates an error margin of 9.7%. Given this rate, the weight estimation error for an adult cow (presuming its actual weight to be 350 kg), would amount to approximately 33.95 kg. There remains a substantial scope for enhancement to align with the practical production demands. Similar to the challenges faced in body dimension measurements, weight regression models established based on body measurements obtained after 3D reconstruction might be affected by individual differences, leading to inaccuracies in body measurement data and subsequently affecting the accuracy of weight estimation. Furthermore, estimating weights using regression equations without considering all body measurement data of the livestock and solely relying on selected major body measurements for weight prediction can be a factor contributing to reduced accuracy in weight estimation results. As summarized in the conclusion of [55], the study extracted a relatively small variety of body measurement data from the surface point clouds of pigs.

### 4.2. Neural Network-Based Visual Weight Estimation

Neural network methods are a novel approach for weight estimation, where CNNs (convolutional neural networks) handle real-time processing of point cloud data. Multi-output regression CNN models are capable of rapidly and accurately extracting the body shape features of pigs and estimating their weight, as depicted in Figure 8.

Kwon et al. [77] reconstructed body measurements from pig point clouds through grid reconstruction to develop a deep neural network (DNN) for weight estimation. They identified 48 measurement types in the grid model and utilized a fully connected deep neural network to estimate weights. The results showed high accuracy in the test dataset with an error of 4.89 kg (relative to the pig's weight error of 2.11%), as shown in Figure 9. He et al. [78] introduced a sheep live weight estimation model based on LiteHRNet (a lightweight high-resolution network). This study used RGB-D images of 726 sheep, and comparative experimental outcomes revealed that the lightweight convolutional neural network (CNN) model trained on RGB-D images achieved acceptable weight estimation results, with an average percentage error (MAPE) of 14.605%. Okinda et al. [79] proposed an image processing feature extraction and adaptive weight estimation neural system, indicating that the system had an average relative error of approximately 3%, with a standard deviation of 0.7%. Dang et al. [80] explored the correlation between body measurements (features) and weight (target values) in cattle using machine learning algorithms. The data analysis results showed high correlations between ten body measurement values and weight. Buayai et al. [81] utilized an artificial neural network to estimate weight. The

experimental evaluation demonstrated the effectiveness and practicality of the method, with an absolute mean error as low as 2.84%. Zhang et al. [73] employed a multi-output regression convolutional neural network (CNN) for pig weight estimation. They modified DenseNet201, ResNet152 V2, Xception, and MobileNet V2 into multi-output regression CNNs, training them on 3D point cloud data, resulting in mean absolute errors (MAEs) for body weight (BW) of 1.16 kg.

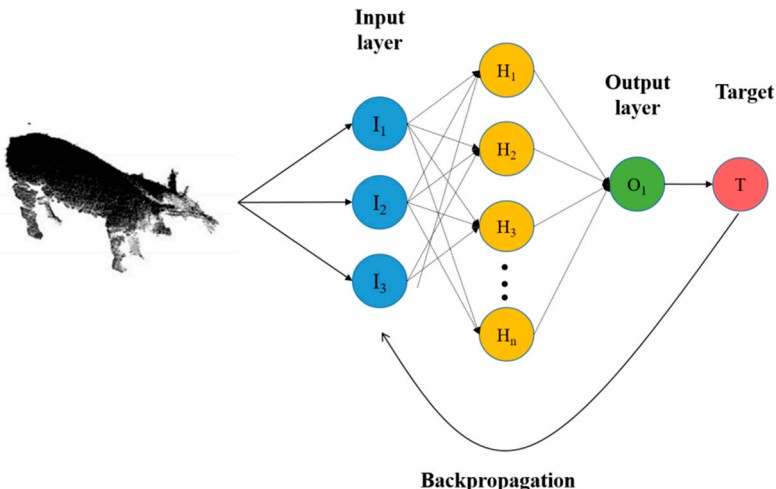

**Figure 8.** Brief framework of neural network-based point cloud weight estimation method.

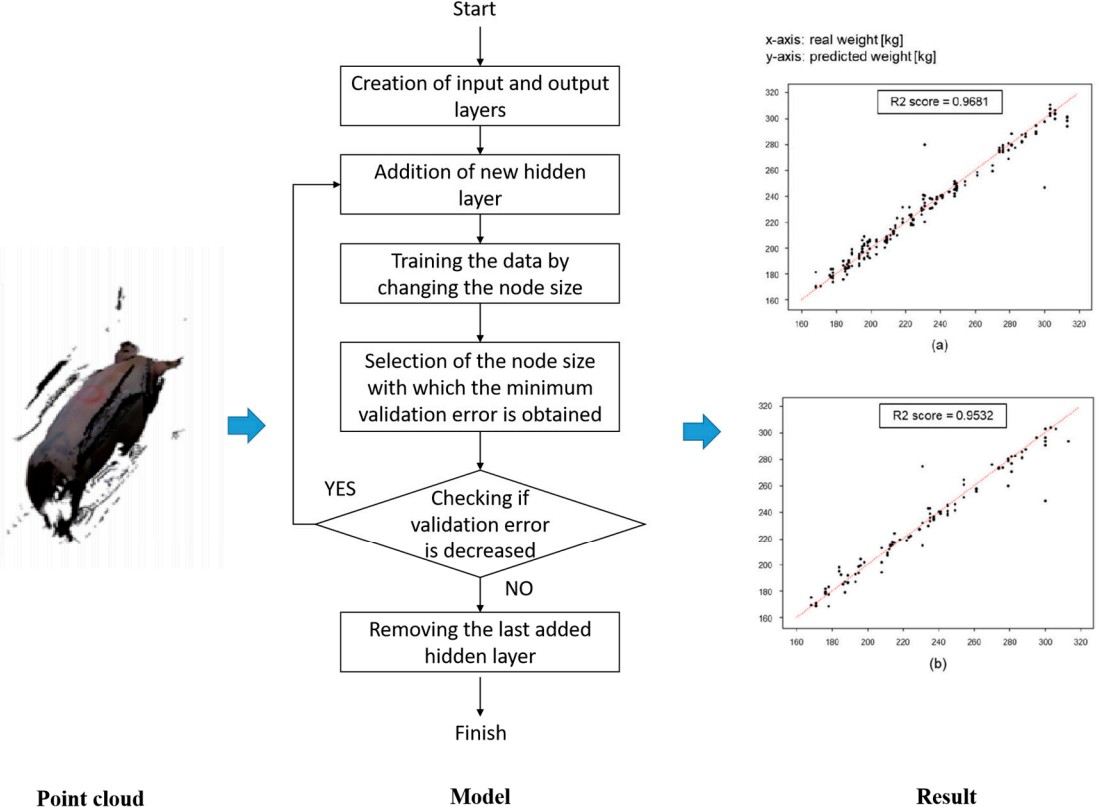

**Figure 9.** Neural network weight estimation process [77].

In conclusion, weight estimation methods based on deep neural networks offer relatively accurate estimations of livestock weight, showcasing robustness, and have become the prevailing research direction presently.

### 4.3. Summary of Weight Estimation Technology

Non-contact livestock weight estimation is a crucial aspect of PLF and is primarily categorized into weight estimation based on the mapping relationship between 3D reconstruction and body measurements and weight estimation utilizing deep convolutional neural networks, as shown in Table 6.

**Table 6.** Comparison of weight estimation with different methods.

| Methods | Advantage | Disadvantage |
|---------|-----------|--------------|
| Estimation of weight based on body dimension | Simple, convenient, and rapid approach, supported by existing research foundations | Relatively lower precision; heavily reliant on the accuracy of 3D reconstruction |
| Weight estimation based on neural network | Strong generalization for diverse livestock batches and types, more accurate precision | Larger data requirements; higher computational resource |

Weight estimation methods based on the mapping relationship between body measurements and weight have successfully achieved non-contact livestock weight estimation with relatively high accuracy as presented in various studies. However, most of these methods rely on a limited set of body measurements such as body width and length, failing to encompass all body measurement information, potentially affecting the model's robustness. Additionally, these methods heavily depend on the accuracy of 3D reconstruction and are susceptible to variations due to individual differences.

To address these issues, recent advancements in neural networks have shown increased accuracy in various neural networks. Researchers have started focusing on applying neural network algorithms to livestock weight estimation, achieving promising progress. This method typically exhibits good generalization capabilities across different batches and breeds of livestock. However, challenges persist. Firstly, training neural network models often necessitates a large volume of labeled data, which could be challenging for livestock weight estimation, especially in scenarios with limited data. Models may tend to overfit, performing well on training data but poorly on new data, necessitating techniques like regularization and data augmentation to mitigate this issue [82]. Secondly, training and deploying neural network models require substantial computational resources, including high-performance hardware and substantial memory, which may not be economically viable for agricultural settings.

## 5. Challenges and Trends

In this paper, we summarized the primary challenges and future research directions in the acquisition of non-contact phenotypic data in livestock based on an extensive literature review and practical research efforts.

### 5.1. The Main Challenges

Currently, there exist the following challenges regarding the non-contact acquisition of phenotypic data in livestock:

(1) Lack of sufficiently accurate 3D reconstruction models: Despite the success of using 3D cameras for point cloud acquisition in recent research, it is crucial to note that non-contact point cloud acquisition of livestock mostly occurs in ideal laboratory settings. However, actual production environments are more complex, with more interference and noise in the point cloud data. Common point cloud registration algorithms may not perform optimally in these conditions. Moreover, variations in lighting conditions in farm environments can result in incomplete point cloud information, limiting accuracy.

(2) Absence of high-quality publicly available livestock point cloud datasets: Massive datasets are crucial for training machine learning or deep learning models in methods related to phenotypic data acquisition and weight estimation. However, due to ownership and confidentiality issues, farms and commercial entities seldom release

their collected data into the public domain. Existing datasets might also exhibit differences in format, structure, or type, necessitating standardization and consistency to ensure data comparability and usability.

(3)  Inefficient point cloud acquisition methods: In many existing studies, the commonly employed channel, suspended, or handheld point cloud acquisition devices result in a single point cloud capture per livestock. Additionally, when using 3D cameras, livestock must remain still for extended periods during data capture. Otherwise, non-rigid deformation could severely affect the reconstruction process, leading to issues like "ghosting". These inefficiencies limit the scalability of these methods in large-scale farming applications.

(4)  Low cost-effectiveness: While acquisition equipment continues to evolve, high-quality 3D cameras still tend to be relatively expensive, which could pose a barrier for some farms and breeding facilities. Furthermore, processing non-contact data often requires complex deep learning models, whose training and fine-tuning might necessitate expensive hardware, potentially reducing cost-effectiveness.

*5.2. Future Development*

With the development of smart sensors, big data, and deep learning, PLF will continue to advance toward non-contact, automated, real-time, and continuous detection while fully considering the survival conditions of livestock and practical production needs. Drawing from the literature review, the following promising future research directions and opportunities are anticipated:

(1)  Enhancement of the accuracy of neural network models: Future developments will involve more machine learning and neural network technologies to handle and analyze large volumes of non-contact data. Improving the accuracy and usability of models can significantly enhance efficiency and make scientifically informed decisions.

(2)  Data integration and standardization: To achieve the sharing and comparison of livestock point cloud data across different breeds and batches, more efforts are needed in data integration and standardization. This will aid in establishing a global livestock phenotypic database, fostering collaboration and development in PLF.

(3)  Efficient data collection: In the future, there will be the use of smaller, lighter, lower-power, and high-resolution 3D cameras for data collection. Additionally, simultaneous data collection from multiple livestock heads could be a direction for improving data collection efficiency.

## 6. Conclusions

Livestock phenotypic data play a crucial role in various aspects of livestock management, breeding, nutrition management, and health monitoring. The acquisition of livestock phenotypic data directly impacts the production efficiency and economic benefits of animal husbandry. However, traditional methods of obtaining phenotypic data are not only time-consuming and labor-intensive but also tend to cause stress to animals, affecting their productivity and welfare. Therefore, these methods are considered inefficient and uneconomical in most farms. This has led to a current lack of livestock phenotypic data. There is a significant amount of literature focusing on researching non-contact methods for acquiring livestock phenotypic data. This article investigated computer vision-based phenotypic data acquisition technologies in precision animal husbandry. Through an extensive review of over 70 relevant studies, we comprehensively analyzed the current research status of livestock 3D reconstruction technology, computer vision-based livestock body dimension acquisition technology, computer vision-based livestock weight estimation technology, and other related aspects. Simultaneously, this discussion delved into the research surrounding these technologies, examining their respective advantages and limitations.

Lastly, we discussed existing challenges, potential future research trends, and opportunities in this field. In the future, we anticipate that automated, real-time neural network models will become the primary direction for acquiring non-contact phenotypic data in

livestock. Network architectures capable of higher precision and utilization of extensive datasets of livestock phenotypic data are expected to become mainstream.

**Author Contributions:** Conceptualization, W.M. and X.Q.; methodology, Y.S.; software, R.G.; validation, L.D. and R.W.; formal analysis, C.P.; investigation, J.Z.; resources, J.W.; data curation, Z.X.; writing—original draft preparation, X.Q.; writing—review and editing, W.M.; visualization, X.Q.; supervision, M.L.; project administration, H.Z.; funding acquisition, S.H. and Q.L. All authors have read and agreed to the published version of the manuscript.

**Funding:** This work was supported by the National Key R&D Program of China (2022ZD0115702), the Beijing Academy of Agriculture and Forestry Sciences (JKZX202214), the Sichuan Science and Technology Program (2021ZDZX0011), the Beijing Nova Program (2022114), the Key Special Project "Promoting Mongolia through Technology" (2022EEDSKJXM012-2), and the Science and Technology Plan Project of Yunnan Provincial Department of Science and Technology (202102AE090039).

**Institutional Review Board Statement:** Ethical review and approval were waived for this study due to it consisting of a literature survey that does not involve humans or animals.

**Data Availability Statement:** The data presented in this study are available upon request from the corresponding author.

**Acknowledgments:** Special thanks to the funding institutions and the people involved.

**Conflicts of Interest:** The authors declare no conflicts of interest.

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
