# Peer review of "Computer Vision-Based Measurement Techniques for Livestock Body Dimension and Weight: A Review"

_agriculture, doi:10.3390/agriculture14020306_

Round 1

Reviewer 1 Report

Comments and Suggestions for Authors

The manuscript is well-written and makes an important contribution to Precision Livestock Farming. However, I recommend a minor revision to improve clarity and conciseness in writing are required.

The main contributions and implications of this manuscript would be agreeably clarified if the authors discussed its complementary.

Comments and Suggestions for Authors

Consider some suggestions below:

Abstract:

Reorganize and structure the abstract to improve the understanding of the scope of your work. Consider including the introduction, the objective, the main points and strengths, and the conclusion categorically. Also consider removing/rewriting redundant terms (e.g.: "This paper provides...; Detailed research...; The article introduces...").

Keywords:

I recommend removing the term "livestock phenotypic acquisition" or substituting it with "Precision Livestock Farming".

Introduction:

P3, L91-97: I suggest rewriting this last paragraph. Could you clarify that they are objectives?

General review:

Could you rearrange the "year" (in ascending order) in some tables?

Improve the quality of the images if possible.

Congratulations on the research, the topic "Challenges and trends" is a very important point of innovation.

Conclusions

Consider rewriting the conclusion of the manuscript including sentences about the methods and forms of acquisition and reconstruction of technology development.

Author Response

Dear Editors and Reviewers:

Thank you for your suggestions and for the reviewers’ comments concerning our manuscript entitled “Computer Vision-Based Measurement Techniques for Livestock Body dimension and Weight: A Survey” (ID: agriculture-2798867). Those comments are all valuable and very helpful for revising and improving our paper, as well as the important guiding significance to our research. We have studied the comments carefully and have made corrections which we hope to meet with approval. Revised portions are marked in yellow in the paper.

The main corrections in this paper and the responses to your comments are as follows:

Responds to the reviewer’s comments:

Comment 1: Reorganize and structure the abstract to improve the understanding of the scope of your work. Consider including the introduction, the objective, the main points and strengths, and the conclusion categorically. Also consider removing/rewriting redundant terms (e.g.: "This paper provides...; Detailed research...; The article introduces...").

Response: Thank you for your generous suggestions. We have reorganized and structured the content in the abstract, eliminating some redundant sentences. We have added introductions to the work, its advantages, objectives, etc., as you mentioned, and also removed some terms. Revised portions are marked in yellow in the paper.

We hope our revisions will receive your approval.

Modified:  Acquiring phenotypic data from livestock constitutes a crucial yet cumbersome phase in the breeding process. Traditionally, obtaining livestock phenotypic data primarily involves manual, on-body measurement methods. This approach not only requires extensive labor but also induces stress on animals, which will lead to potential economic losses. Presently, the integration of next-generation Artificial Intelligence (AI), visual processing, intelligent sensing, multimodal fusion processing, and robotic technology is increasingly prevalent in livestock farming. The advantages of these technologies lie in their rapidity and efficiency, coupled with their capability to acquire livestock data non-contactly. Based on this, we provides a comprehensive summary and analysis of the primary advanced technologies employed in the non-contact acquisition of livestock phenotypic data. It focuses on visual and AI-related techniques, including 3D reconstruction technology, body dimension acquisition techniques, and live animal weight estimation. We introduces the development of livestock 3D reconstruction technology and compares the methods of obtaining 3D point cloud data of livestock through RGB cameras, laser scanning, and 3D cameras. Subsequently, we explores the body size calculation methods and compares the advantages and disadvantages of RGB image calculation methods and 3D point cloud body size calculation methods. Furthermore, we also compares and analyzes weight estimation methods of linear regression and neural networks. Finally, it discusses the challenges and future trends of non-contact livestock phenotypic data acquisition. Through emerging technologies like next-generation AI and computer vision, the acquisition, analysis, and management of livestock phenotypic data are poised for rapid advancement.(lines14-33)

Comment 2: I recommend removing the term "livestock phenotypic acquisition" or substituting it with "Precision Livestock Farming".

Response: Thank you for your generous suggestions.We also believe that 'Precision Livestock Farming' is more suitable as a keyword, and we have made corrections accordingly.

Comment 3: P3, L91-97: I suggest rewriting this last paragraph. Could you clarify that they are objectives?

Response: Thank you for your generous suggestions.We apologize for the lack of clarity in our manuscript. We have rewritten this section to more closely align with the summary of the article's work.

Modified

In this paper, our focus lies on non-contact intelligent sensing technology for livestock. Specifically, we emphasize intelligent perception and analysis techniques related to three primary tasks: 1) computer vision-based 3D reconstruction of livestock; 2) computer vision-based livestock body dimension acquisition technology; 3) computer vision-based livestock weight estimation technology. Within this work, we summarize and analyze the latest advancements in these fields and discuss future research opportunities, as well as the associated challenges.(lines90-96)

Comment 4: Could you rearrange the "year" (in ascending order) in some tables?

Improve the quality of the images if possible.

Response: Thank you for your valuable suggestions; we have adjusted the tables accordingly. Due to the different livestock subjects measured in Table 5, we have reorganized the Table by classifying the livestock and then sorting it in ascending order by year.We are sorry for the oversight that led to the distortion of the figure. We have remade the figure. 

Modified: (lines 119,lines 163 and lines 423)We remade the figure in lines340 and lines 492.

Comment 5: Consider rewriting the conclusion of the manuscript including sentences about the methods and forms of acquisition and reconstruction of technology development.

Response: Thank you for your suggestion. We have rewritten and supplemented the sentences concerning the methods and forms of acquisition and reconstruction technology development, hoping to gain your approval.

Modified: (lines588-591)

We tried our best to improve the manuscript and made some changes in the manuscript. These changes will not influence the content and framework of the paper.

In all, we found the reviewer’s comments are quite helpful, and I revised my paper point-by-point. We appreciate for Editors and Reviewers’ warm work earnestly, and hope that the correction will meet approval. Once again, thank you very much for your comments and suggestions.

Best wishes.

All authors

Reviewer 2 Report

Comments and Suggestions for Authors

Title: Computer Vision-Based Measurement Techniques for Livestock Body Dimension and Weight: A Survey.

The manuscript critically reviews the computer-based techniques to predict weight using body dimensions. I believe the manuscript is a critical review, not a survey. My suggestion is to change the title.

The manuscript is well structured; however, the summary of each section lacks a deep discussion with references to the pointed critics. For instance, in lines 437 to 440, you did not deeply comment on the previously found accuracies and blamed the lack of weight accuracy when the method was applied. Your analysis in the previous research looks inappropriate since you did not focus specifically on their accuracy. I suggest you review the overall discussion based on the items you analyzed regarding the selected topics and references.

Another point that needs correction is the figures that seem pretty distorted. Please see the attached file for more corrections.

Comments on the Quality of English Language

Few corrections are needed.

Author Response

Responds to the reviewers’ comments

Dear Editors and Reviewers:

Thank you for your suggestions and for the reviewers’ comments concerning our manuscript entitled “Computer Vision-Based Measurement Techniques for Livestock Body dimension and Weight: A Survey” (ID: agriculture-2798867). Those comments are all valuable and very helpful for revising and improving our paper, as well as the important guiding significance to our research. We have studied the comments carefully and have made corrections which we hope to meet with approval. Revised portions are marked in yellow in the paper.

The main corrections in this paper and the responses to your comments are as follows:

Responds to the reviewer’s comments:

Reviewer #1:

Comment1: The manuscript critically reviews the computer-based techniques to predict weight using body dimensions. I believe the manuscript is a critical review, not a survey. My suggestion is to change the title. 

Response: Thank you for your valuable suggestion. Indeed, this article is a review, not a survey. Thank you for pointing out the error. 

Modified: We rewrite the title as follows:

Computer Vision-Based Measurement Techniques for Livestock Body dimension and Weight: A Review

Comment2: the summary of each section lacks a deep discussion with references to the pointed critics. For instance, in lines 437 to 440, you did not deeply comment on the previously found accuracies and blamed the lack of weight accuracy when the method was applied. Your analysis in the previous research looks inappropriate since you did not focus specifically on their accuracy. I suggest you review the overall discussion based on the items you analyzed regarding the selected topics and references.

Response: Thank you for your valuable suggestion. We are sorry that we neglected to  the deep discussion with references to the pointed critics. This indeed affects the quality of the article's discussion. This is our mistake. Therefore, according to your suggestion, we add a discussion in lines 447-452,459-461; lines348; lines142-144; line291 and so on.

Modified: 

As presented in document [38], the reconstructed area only reached 60%, which obviously does not meet the requirements for subsequent body size and weight calculations.(lines142-144).

As shown in document [73], the error reached 10.2%. (lines291)

The aforementioned studies have achieved the estimation of livestock weight. However, their accuracy is relatively low. For instance, the study referenced as [60] indicates an error margin of 9.7%. Given this rate, the weight estimation error for an adult cow(presuming its actual weight to be 350 kg), would amount to approximately 33.95 kg. There remains a substantial scope for enhancement to align with the practical production demands. (lines447-452)

Comment3: Another point that needs correction is the figures that seem pretty distorted. Please see the attached file for more corrections.

Response: We are sorry for the oversight that led to the distortion of the figure. We have remade the figure. 

And one of the figures was displayed in the original paper; it is the original image.We are sorry for we can not remade that figure.

Modified: We remade the figure in lines340 and lines 492.

but the figure(a) which in lines 382 is the original figure.

Please allow me to explain the issue in Figure 3. In Figure 3(a), the image above represents the data acquisition method, which involves the use of a collection system equipped with LiDAR sanning and a camera--Morpho3D, as described in literature[32].The Figure3(b) below is the 3D reconstruction result from literature [32].

We tried our best to improve the manuscript and made some changes in the manuscript. These changes will not influence the content and framework of the paper.

In all, we found the reviewer’s comments are quite helpful, and I revised my paper point-by-point. We appreciate for Editors and Reviewers’ warm work earnestly, and hope that the correction will meet approval. Once again, thank you very much for your comments and suggestions.

Best wishes.

All authors

Reviewer 3 Report

Comments and Suggestions for Authors

This manuscript is well developed and well written. A quick review for grammar is needed and the References must be reviewed for consistency. 

Comments on the Quality of English Language

Well Written

Author Response

Dear Editors and Reviewers:

Thank you for your suggestions and for the reviewers’ comments concerning our manuscript entitled “Computer Vision-Based Measurement Techniques for Livestock Body dimension and Weight: A Survey” (ID: agriculture-2798867). Those comments are all valuable and very helpful for revising and improving our paper, as well as the important guiding significance to our research. We have studied the comments carefully and have made corrections which we hope to meet with approval. Revised portions are marked in yellow in the paper.

The main corrections in this paper and the responses to your comments are as follows:

Responds to the reviewer’s comments:

We have revisited the references section and made revisions.

We tried our best to improve the manuscript and made some changes in the manuscript. These changes will not influence the content and framework of the paper.

In all, we found the reviewer’s comments are quite helpful, and I revised my paper point-by-point. We appreciate for Editors and Reviewers’ warm work earnestly, and hope that the correction will meet approval. Once again, thank you very much for your comments and suggestions.

Best wishes.

All authors

Round 2

Reviewer 2 Report

Comments and Suggestions for Authors

The authors reviewed the manuscript according the the reviewer suggestions.